# Holocentromeres are dispersed point centromeres localized at transcription factor hotspots

**Florian A Steiner, Steven Henikoff\***

Basic Sciences Division, Howard Hughes Medical Institute, Fred Hutchinson Cancer Research Center, Seattle, United States

**Abstract** Centromeres vary greatly in size and sequence composition, ranging from 'point' centromeres with a single cenH3-containing nucleosome to 'regional' centromeres embedded in tandemly repeated sequences to holocentromeres that extend along the length of entire chromosomes. Point centromeres are defined by sequence, whereas regional and holocentromeres are epigenetically defined by the location of cenH3-containing nucleosomes. In this study, we show that *Caenorhabditis elegans* holocentromeres are organized as dispersed but discretely localized point centromeres, each forming a single cenH3-containing nucleosome. These centromeric sites co-localize with kinetochore components, and their occupancy is dependent on the cenH3 loading machinery. These sites coincide with non-specific binding sites for multiple transcription factors ('HOT' sites), which become occupied when cenH3 is lost. Our results show that the point centromere is the basic unit of holocentric organization in support of the classical polycentric model for holocentromeres, and provide a mechanistic basis for understanding how centromeric chromatin might be maintained.

**\*For correspondence:** steveh@fhcrc.org

**Reviewing editor**: Asifa Akhtar, Max Planck Institute for Immunobiology and Epigenetics, Germany

## Introduction

The centromere is a defining feature of eukaryotic chromosomes and is essential for the segregation of chromosomes during cell division, as it organizes the proteinaceous kinetochore for attachment to the spindle apparatus at mitosis. Centromeres are universally marked by the variant histone cenH3 (also called CENP-A in many organisms) that replaces canonical histone H3 in centromeric nucleosomes, and most commonly localize to a single position along the chromosome (*Malik and Henikoff, 2009*). However, the DNA on which centromeric nucleosomes assemble is not conserved and varies greatly in size and composition. It ranges from genetically defined point centromeres that assemble a single cenH3-containing nucleosome to epigenetically defined regional centromeres of several kb or Mb of tandemly repeated DNA to holocentromeres that extend along the length of entire chromosomes. With the exception of budding yeast point centromeres, where there is a 1:1 relationship between a single cenH3 nucleosome and the functional centromere, the precise organization of centromeric chromatin has remained elusive. One of the main issues standing in the way of uncovering the distribution of centromeric nucleosomes is the fact that most regional centromeres are localized to homogeneous tandemly repetitive regions of the genome, making it difficult to map individual nucleosomes. *C. elegans* is amenable to address this question because the genome is repeat-poor, making it possible to precisely map centromeric regions.

Classical cytogenetic observations have demonstrated that *C. elegans* chromosomes are holocentric, whereby mitotic spindle fibers attach along the length of chromosomes and pull them to the poles as straight bars rather than from a single position that defines the more familiar monocentric chromosomes (*Schrader, 1935*; *Albertson and Thomson, 1982*). Two models have been put forward for how

**eLife digest** During cell division, the chromosomes in the original cell must be replicated and these 'sister chromosomes' must then be divided equally between the two new daughter cells. At first, the sister chromosomes are held together near a region called the centromere, which is important because the microtubules that pull the sister chromosomes apart attach themselves to the centromere. In many cases, the centromere is a small region near the middle of the chromosomes, which produces a classic X shape. However, in some organisms centromeres span the entire length of the chromosomes. There are at least 13 plant and animal lineages with such holocentromeres.

Inside the nucleus of cells, DNA is wrapped around molecules called histones. There are five major families of histones, and histones belonging to one of these families—the H3 histones—are replaced by cenH3 variant histones at both conventional centromeres and holocentromeres. There are many unanswered questions about holocentromeres. In particular, do holocentromeres truly extend along the full length of the chromosomes, or are they found at a large number of specific sites?

Now Steiner and Henikoff have studied the distribution of cenH3 in the genome of the worm *C. elegans* to investigate holocentromeres in greater detail. These experiments showed that the holocentromere in *C. elegans* is actually made of about 700 individual centromeric sites distributed along the length of the chromosomes. Each of these sites contains just one nucleosome that contains cenH3, and these sites are likely to be the sites that microtubules attach to during cell division. Surprisingly, the same sites can also act as so-called 'HOT–sites': these sites are bound by many proteins that are involved in regulating the process by which genes are expressed as proteins, which suggests a link between centromeres and these regulatory proteins.

The work of Steiner and Henikoff describes how centromeric nucleosomes are distributed across the genome, but why and how cenH3 ends up at these particular 700 sites remains an open question.

holocentric chromosomes might be organized (*Schrader, 1947*). The 'diffuse centromere' model predicts that the centromere is truly distributed along the length of the chromosomes, and that spindle fiber attachments form randomly. The 'polycentromere' model predicts that there are a number of discrete sites dispersed along the chromosomes, creating a holocentric appearance when observed at cytological resolution (*Figure 1A*).

Consistent with either model, *C. elegans* cenH3 (also called HCP-3) localizes to a characteristic band along the length of the chromosome at mitosis (*Buchwitz et al., 1999*). A previous study mapped *C. elegans* cenH3 using a microarray-based approach and found that it occupies ~2900 broad domains that account for about half of the genome, but that there was only enough cenH3 to cover 4% of the genome, suggesting that cenH3 nucleosomes assemble at random positions within the domains (*Gassmann et al., 2012*). These findings thus seemingly supported the diffuse centromere model. However, mitotic microtubules must attach to discrete sites for chromosome segregation, and the number of microtubules attached during *C. elegans* mitosis has been estimated to about 100 for all six chromosomes combined (*O'Toole et al., 2003*). This left open the question of the relationship between these diffuse domains and discrete microtubule attachment sites.

To identify potential kinetochore attachment sites in *C. elegans*, we profiled cenH3 nucleosomes with single base-pair resolution. While we observed domains of low occupancy similar to those described in the earlier study, we also discovered discrete sites of much higher cenH3 occupancy that are distributed independently of the domains. Depletion of the machinery needed for incorporation of cenH3 nucleosomes resulted in reduced occupancy of cenH3 at these sites. As an independent indicator of centromeric localization, we also profiled the inner kinetochore protein CENP-C (also called HCP-4 in *C. elegans*). We found that cenH3 sites coincide with high CENP-C signal, indicating that they serve as attachment sites for the kinetochore, consistent with a polycentric organization of the chromosome. Individual sites resemble budding yeast point centromeres and coincide with transcription factor hotspots, which become occupied by transcription factors when cenH3 is lost, providing a clue as to how kinetochore sites might be selected and maintained.

**A**   Diffuse holocentromere      Polycentric holocentromere

- Centromere
- Chromatin

**B**

- CenH3 X-ChIP-chip
- CenH3 N-ChIP-seq
- H3.3 N-ChIP-chip
- H3K9me3 X-ChIP-chip

Chr I   5.7   6.1 Mb

log2(ChIP/input)

**C**

5673   5675   5936   5938 kb   Chr I

log2(ChIP/input)

- CenH3 X-ChIP-chip
- CenH3 N-ChIP-seq

**D**

Chr I   Chr II   Chr III   Chr IV   Chr V   Chr X

106   117   106   121   95   162

Total: 707

**E**

Input      CenH3 N-ChIP

high cenH3 signal      low

Normalized counts

Distance from cenH3 sites (kb)

**Figure 1**. Genome-wide distribution of cenH3. (**A**) Classic holocentromere models proposed by Schrader (*Schrader, 1947*): diffuse and polycentric holocentromeres. The diffuse model predicts full centromere coverage of the chromosomes. The polycentric model predicts discrete centromeric sites that together give the appearance of holocenticity. (**B**) Genome browser view of 525 kb on Chr I for cenH3 X-ChIP-chip (*Gassmann et al., 2012*), cenH3

*Figure 1. Continued on next page*

*Figure 1. Continued*

N-ChIP-seq (this study), H3.3 N-ChIP-chip (***Ooi et al., 2010***) and H3K9me3 X-ChIP-chip (***Liu et al., 2011***), showing the close correspondance of cenH3 N-ChIP and X-ChIP signals for domains, but not peaks, and the positive correlation of cenH3 signal with H3K9me3 signal and the negative correlation of cenH3 signal with H3.3 signal. $Log_2$ ratios of IP and input are shown to enable comparison between microarray and sequencing data. CenH3 peaks are marked by asterisks. (**C**) Genome browser view of cenH3 N-ChIP-seq (this study) and cenH3 X-ChIP-chip (***Gassmann et al., 2012***) at two representative cenH3 peaks marked in (**B**) with red boxes. (**D**) Number and genomic distribution of cenH3 peaks called per chromosome. See also ***Figure 1—source data 1***. (**E**) Heatmaps and average plots of input and cenH3 N-ChIP signal within a 2-kb window around all 707 cenH3 peaks. Each line of the heatmaps represents an individual cenH3 site. The heatmaps are sorted from high to low cenH3 signal.

The following source data and figure supplements are available for figure 1:

**Source data 1**. Genomic coordinates of cenH3 peaks.

**Figure supplement 1**. Solubilization of chromatin.

**Figure supplement 2**. Comparison of cenH3 to H3K9me3 and H3.3 on a genome-wide scale.

**Figure supplement 3**. Analysis of cenH3 distribution and occupancy on a genome-wide scale.

## Results

### CenH3 levels are high in domains of low nucleosome turnover

To precisely localize cenH3-containing nucleosomes and identify potential kinetochore attachment sites, we digested chromatin from mixed-stage embryos with micrococcal nuclease (MNase) and solubilized the majority of the chromatin by cavitation, a method adapted from ***Jin and Felsenfeld, 2007*** (***Figure 1—figure supplement 1***). We subsequently performed native chromatin immunoprecipitation (ChIP) of cenH3 from the soluble chromatin, followed by paired-end sequencing (N-ChIP-seq), resulting in single base-pair resolution maps of cenH3-associated DNA fragments.

As expected from a previous ChIP-microarray map of *C. elegans* cenH3 using formaldehyde crosslinking (X-ChIP-chip) (***Gassmann et al., 2012***), we found that cenH3 is broadly distributed throughout the genome. CenH3 is enriched towards the arms relative to the centers of the autosomes, while the distribution on the X chromosome is relatively even (***Figure 1—figure supplement 2A***). In *C. elegans*, chromosome arms tend to be enriched for repeats and are associated with marks of heterochromatin (***The C. elegans Sequencing Consortium, 1998***; ***Liu et al., 2011***). Indeed, the distribution of cenH3 is positively correlated with the distribution of trimethylation of lysine 9 on histone H3 (H3K9me3), a mark of transcriptionally silent regions, in both our ChIP-seq and the previously published ChIP–chip data (***Figure 1—figure supplement 2B***, left panels. r = 0.44, p<2.2 × 10⁻¹⁶ for correlation with cenH3 X-Chip and r = 0.31, p<2.2 × 10⁻¹⁶ for correlation with N-ChIP) (***Gu and Fire, 2010***; ***Liu et al., 2011***). This is consistent with previous findings that have associated H3K9 methylation with the acquisition of cenH3 in fission yeast (***Folco et al., 2008***; ***Kagansky et al., 2009***). We therefore wondered if the enrichment of cenH3 is associated with lower nucleosome turnover. The replication-independent variant histone H3.3 is incorporated into chromatin when nucleosomes are replaced and serves as a measure of replication-independent nucleosome turnover (***Ahmad and Henikoff, 2002***; ***Mito et al., 2005***; ***Goldberg et al., 2010***; ***Ooi et al., 2010***). Consistent with the hypothesis that cenH3 is associated with lower nucleosome turnover, we found that the distributions of H3.3 and cenH3 are negatively correlated in both our ChIP-seq and the previously published ChIP–chip data (***Figure 1—figure supplement 2B***, center panels. r = −0.65, p<2.2 × 10⁻¹⁶ for correlation with cenH3 X-Chip and r = −0.21, p<2.2 × 10⁻¹⁶ for correlation with N-ChIP). Replication-independent nucleosome turnover is mainly driven by transcription (***Deal et al., 2010***; ***Teves and Henikoff, 2011***), consistent with the previously described anti-correlation between cenH3 and RNA polymerase II (***Gassmann et al., 2012***).

CenH3 has been found to be localized to 10–12 kb wide domains that occupy about half of the genome (***Figure 1B***, first track) (***Gassmann et al., 2012***). Despite the use of a different methodology (low-salt native chromatin preparation and MNase digestion instead of formaldehyde fixation and sonication, a different antibody and ChIP-seq instead of ChIP–chip), our data showed a very similar domain pattern (***Figure 1B***, second track, ***Figure 1—figure supplement 2B***, right panel. Correlation with cenH3 X-ChIP r = 0.67, p<2.2 × 10⁻¹⁶). The previously published data (***Gassmann et al., 2012***) pointed to an

anti-correlation of cenH3 occupancy with transcription in the germline and in the early embryo. Because there is little RNA Polymerase II activity in the early embryo, and transcriptional profiles are not directly transmitted during the maternal–zygotic transition, the authors proposed that it is the memory of germline transcription transmitted to the embryo that excludes cenH3 incorporation. We found that the domains were both negatively correlated with previously published H3.3 data (*Figure 1B*, third track) and positively correlated with previously published H3K9me3 patterns (*Figure 1B*, fourth track) (*Ooi et al., 2010*; *Liu et al., 2011*). H3.3 is abundantly incorporated into chromatin in the germline and throughout early embryonic development (*Ooi et al., 2010*), whereas H3K9me3 is associated with transcriptionally silent chromatin where nucleosome turnover and H3.3 incorporation are low (*Ahmad and Henikoff, 2002*; *Mito et al., 2005*; *Gu and Fire, 2010*; *Ooi et al., 2010*; *Liu et al., 2011*). This suggests that it is nucleosome turnover that excludes the deposition of cenH3 and shapes the domain-like distribution of cenH3 across the genome, which can happen even in absence of transcription in the early embryo.

## High-resolution mapping of cenH3 reveals discrete high-occupancy sites

The levels of cenH3 within a cell only allow for the occupancy of 4% of the genome, and each cenH3 domain can therefore only contain a limited number of cenH3 nucleosomes per cell (*Gassmann et al., 2012*). The large-scale correspondence of our native ChIP-seq data to the previously published crosslinked ChIP–chip data provides independent confirmation of this interpretation. However, we wondered whether there might also be preferred sites of centromeric nucleosome positioning within the cenH3 domains of *C. elegans*, as predicted by the polycentromere model. These sites would appear as sites of high cenH3 occupancy in a population average. We indeed found that cenH3 was highly enriched at discrete, dispersed loci (*Figure 1B,C*). As these loci appeared as very well-defined peaks, we removed background by subtracting the input signal and considered sites with 30 or more normalized counts (equivalent to the mean plus 7 times the standard deviation of the genome-wide signal) in at least one of two biological replicates as positive. We identified about 100 cenH3 peaks on each chromosome (707 peaks total; *Figure 1D*), with the distance between peaks ranging from 290 bp to 1.9 Mb (median 83 kb; *Figure 1—figure supplement 3A*, *Figure 1—source data 1*). We averaged the signal around all 707 cenH3 peaks, represented by a single centered peak in the cenH3 ChIP data that is highly enriched compared to input (*Figure 1E*). The peaks were enriched in gene-poor regions of the genome (*Figure 1—figure supplement 3B*). To our surprise, 607 out of 707 of these peaks resided outside of the domains described previously (*Gassmann et al., 2012*) and corresponded to sites of only slight local cenH3 enrichment in the X-ChIP data (*Figure 1—figure supplement 3C*). In the N-ChIP data, cenH3 occupancy was much higher at these peaks compared to the domains (*Figure 1—figure supplement 3D*). We hypothesized that these sites are preferred for the deposition of centromeric nucleosomes and serve as potential kinetochore attachment sites.

## cenH3 peaks are hyper-sensitive to MNase digestion

Previous studies in other organisms observed that centromeric regions were sensitive to MNase or resulted in patterns inconsistent with the presence of canonical nucleosomes (*Polizzi and Clarke, 1991*; *Takahashi et al., 1992*; *Dalal et al., 2007*; *Krassovsky et al., 2012*). We found that the cenH3 peaks were sensitive to MNase and disappeared with progressive MNase digestion (*Figure 2A,B*), even at MNase conditions where nucleosome arrays remain intact and that would be considered underdigested by most standards (*Figure 2—figure supplement 1*, left panel). In contrast, the chromatin features around the cenH3 peaks were remarkably unaffected (*Figure 2A*, *Figure 2—figure supplement 2*). As a control, we compared the cenH3 peaks to the +1 nucleosomes at transcription start sites (*Chen et al., 2013*). Occupancy of these well-positioned nucleosomes also decreased with progressing MNase digestion, but to a lesser extent (*Figure 2C*). We quantified MNase sensitivity at cenH3 peaks, at the nucleosomes immediately flanking the cenH3 peaks, and at the +1 nucleosomes by dividing the occupancy of these features at each time point by the occupancy at the first time point. This analysis revealed that cenH3 nucleosomes were more sensitive to MNase than both flanking and +1 nucleosomes (*Figure 2D*). These findings suggest that the sites of high cenH3 occupancy contain nucleosomes with similar properties as centromeric nucleosomes in other organisms.

## CenH3 site occupancy depends on cenH3 loading

Incorporation of cenH3 into chromatin depends on the kinetochore protein KNL-2 (*Maddox et al., 2007*). To test if the signal at the cenH3 peaks results from KNL-2-dependent incorporation of cenH3,

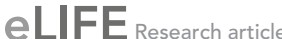

**Figure 2**. CenH3 peaks are especially MNase sensitive. (**A**) Genome browser view of input chromatin and cenH3 ChIP signal within a 25-kb window surrounding two representative cenH3 peaks. Tracks for occupancy after 1 min, 2 min, 5 min and 10 min of MNase digestion are shown. (**B**) Average input signal (left) and cenH3 ChIP signal (right) within a 1-kb window around all 707 cenH3 sites after the indicated MNase digestion intervals. The
*Figure 2. Continued on next page*

*Figure 2. Continued*
dashed red line indicates the midpoint of the cenH3 nucleosome and the dashed black lines indicate the midpoints of the flanking nucleosomes.
(**C**) Average input signal within a 1-kb window around 7043 transcriptional start sites (TSS) after the indicated MNase digestion intervals. The dashed green line indicates the midpoint of the +1 nucleosome. TSS were defined by Chen et al. (***Chen et al., 2013***). (**D**) MNase sensitivity plot for the cenH3 nucleosome and the flanking nucleosomes shown in (**B**) and the +1 nucleosome shown in (**C**). The occupancy of these nucleosomes at each MNase digestion time point was divided by the occupancy at the first time point. N = 707 (cenH3 nuc), 1414 (flanking nucs), 7043 (+1 nuc).

The following figure supplements are available for figure 2:

**Figure supplement 1**. Progress of MNase digestion.

**Figure supplement 2**. CenH3 sites are preferentially digested by MNase.

we analyzed the chromatin upon KNL-2 knockdown. KNL-2 depletion by RNAi led to an embryonic lethal phenotype with 99 ± 1% penetrance (n = 8). We confirmed by microscopy that this was caused by chromosome segregation defects and found by immunofluorescence that the cenH3 signal became undetectable in embryos. These results were consistent with published findings (***Maddox et al., 2007***) and suggested that our depletion of KNL-2 successfully reduced the presence of cenH3 and thus the functionality of centromeres. ChIP experiments revealed that cenH3 occupancy at the cenH3 sites was much reduced in *knl-2(RNAi)* embryos compared to wildtype (***Figure 3***) for similar levels of MNase digestion (***Figure 2—figure supplement 1***, center panel). This effect extended genome wide (***Figure 3—figure supplement 1A***) and was not caused by changes in the input chromatin, as the overall occupancy and positioning of most canonical nucleosomes and other DNA binding factors remained unchanged in *knl-2(RNAi)* embryos (***Figure 3—figure supplement 1B***), and the input chromatin showed the same correlation with wildtype chromatin as between wildtype replicates (R = 0.971 and 0.969 for wildtype vs *knl-2(RNAi)* and R = 0.973 for wildtype vs wildtype; comparison of normalized fragment counts in 10-bp bins, N = 7633808).

It is conceivable that with a partial depletion of the factor required for cenH3-assembly into chromatin, relatively high cenH3 occupancy is maintained at the sites that are functional due to perdurance of the protein. Indeed, cenH3 is still locally enriched at cenH3 sites in the knockdown, albeit at much reduced levels, while the broad domains of weak enrichment are lost (***Figure 3A***, ***Figure 3—figure supplement 1***). These results indicate that the signal at the identified cenH3 sites indeed depends on the incorporation of cenH3 into centromeric nucleosomes by the cenH3-specific assembly machinery.

## CenH3 peaks correspond to kinetochore sites

CenH3 can be incorporated at low levels into nucleosomes away from the centromeres (***Camahort et al., 2009***; ***Lefrancois et al., 2009***; ***Lopes da Rosa et al., 2011***; ***Krassovsky et al., 2012***; ***Lefrancois et al., 2013***; ***Lacoste et al., 2014***), and so the presence of cenH3 itself is thus not a sufficient measure for the presence of a centromere. To test if the cenH3 peaks indeed correspond to centromeric sites, we compared it to the kinetochore. Previous findings from our lab have suggested that in budding yeast the chromatin fraction that remains insoluble under native conditions after MNase digestion is strongly enriched for kinetochore complexes (***Krassovsky et al., 2012***). As there is a one-to-one relationship between the centromere and the kinetochore in budding yeast, and the kinetochore components are conserved between eukaryotes, it can be inferred that kinetochore-bound chromatin remains mostly insoluble under these conditions. We therefore analyzed the distribution of the MNase fragments associated with the chromatin fraction that remained insoluble after MNase digestion and needle extraction and found peaks corresponding to each cenH3 peak (***Figure 4A,B***). This suggested the presence of insoluble complexes, potentially kinetochores, at every cenH3 site in at least part of the cell population analyzed. The insoluble chromatin signal is reduced in *knl-2(RNAi)* embryos (***Figure 4—figure supplement 1A,B***), supporting the interpretation that these peaks correspond to kinetochores. Interestingly, the peaks in the insoluble chromatin were more resistant to MNase than cenH3 nucleosomes in the soluble fraction, and the insoluble chromatin signal persisted even after 10 min of MNase digestion (***Figure 4A,B***), suggesting that the proteins that render cenH3 chromatin insoluble during extraction help to protect it from nuclease digestion.

To test directly if the cenH3 peaks coincide with kinetochore attachment sites, we performed ChIP on the inner kinetochore protein CENP-C. This protein binds cenH3 and is required for the assembly



**Figure 3**. CenH3 peaks depend on cenH3 loading. (**A**) Genome browser view of cenH3 ChIP in wildtype and *knl-2(RNAi)* embryos, with two enlarged cenH3 peaks. KNL-2 is required for cenH3 loading onto chromatin. Differences between ChIP and input are shown. CenH3 peaks are marked by asterisks. (**B**) Average cenH3 ChIP signal within a 2-kb window around all 707 cenH3 sites in wildtype and *knl-2(RNAi)* embryos. Differences between ChIP and input are plotted. (**C**) Heatmap of difference in cenH3 ChIP signal between wildtype and *knl-2(RNAi)*. Each line of the heatmap represents an individual cenH3 site. The heatmap is sorted by decreasing difference between wildtype and *knl-2(RNAi)*.

The following figure supplements are available for figure 3:

**Figure supplement 1**. Comparison of wildtype and *knl-2(RNAi)* chromatin.

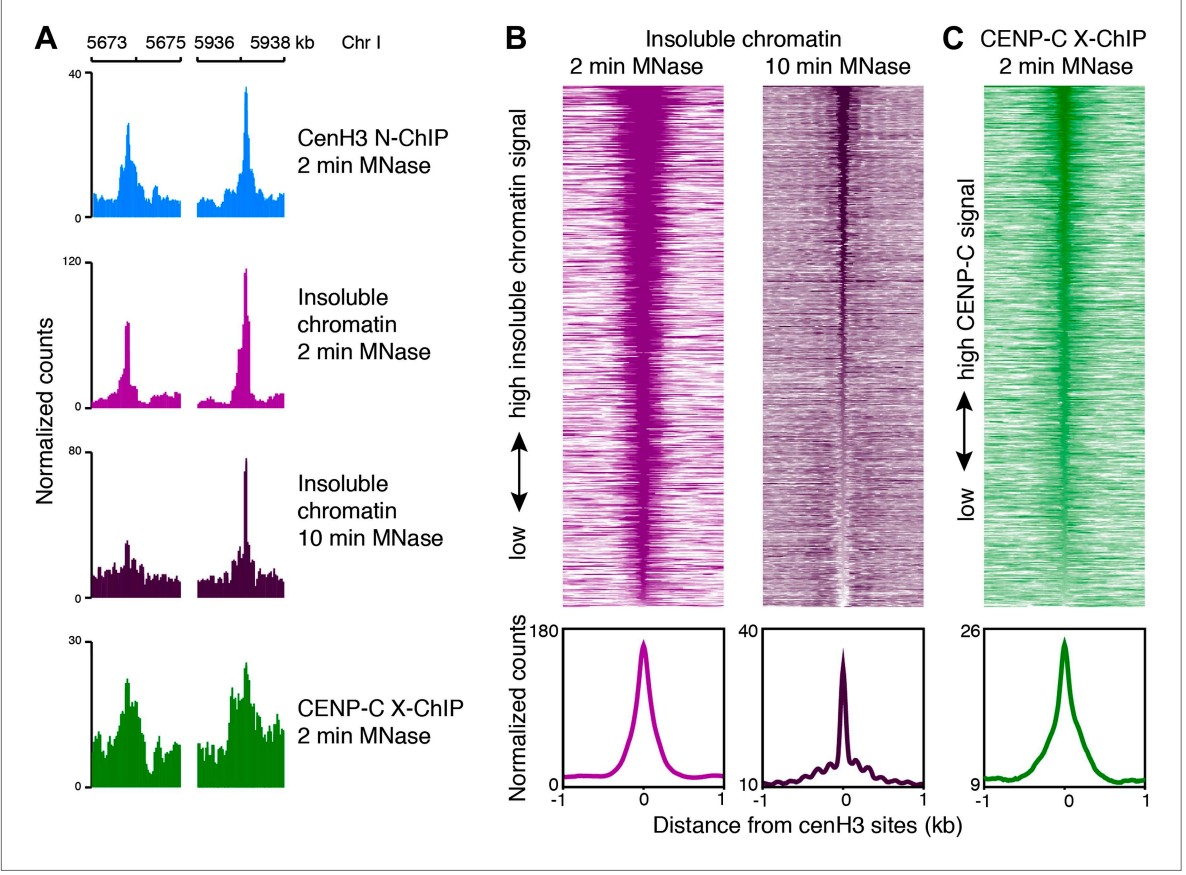

**Figure 4**. CenH3 sites are bound by the kinetochore. (**A**) Genome browser view of cenH3 ChIP (native, 2 min MNase), insoluble chromatin (native, 2 min and 10 min MNase), and CENP-C ChIP (formaldehyde-crosslinked, 2 min MNase) signal at two representative cenH3 sites. (**B** and **C**) Heatmaps and average plots of insoluble chromatin signal after 2-min and 10-min MNase digestion (**B**) and CENP-C ChIP signal (**C**) within a 2-kb window around all 707 cenH3 sites. Each line of the heatmaps represents an individual cenH3 site. Heatmaps are sorted by decreasing signal.

The following figure supplements are available for figure 4:

**Figure supplement 1**. Insoluble chromatin at centromeric sites in *knl-2(RNAi)* embryos.

**Figure supplement 2**. Quantification of kinetochore occupancy.

of the kinetochore complex that links centromeric chromatin to the microtubule (*Moore and Roth, 2001*; *Oegema et al., 2001*; *Cheeseman et al., 2004*; *Carroll et al., 2010*; *Kato et al., 2013*). In fact, CENP-C can organize the entire functional kinetochore in the absence of cenH3 (*Gascoigne et al., 2011*; *Przewloka et al., 2011*; *Hori et al., 2013*). Although CENP-C remains with centromeric DNA throughout the cell cycle in other organisms, in *C. elegans* CENP-C localizes to chromatin only during mitosis, but not interphase, when it is in the cytoplasm (*Moore and Roth, 2001*). As a consequence, only a fraction of the cells analyzed contains CENP-C on chromatin, which limits the dynamic range of ChIP signal that is achievable. Because the kinetochore complex is highly insoluble, no DNA was recovered in native ChIP of CENP-C (data not shown). We therefore profiled CENP-C using MNase followed by formaldehyde crosslinking and solubilization with SDS. We found that CENP-C is enriched at cenH3 sites (*Figure 4A,C*). Neither the signal in the insoluble fraction nor the signal for CENP-C was enriched over the previously identified cenH3 domains compared to the rest of the genome (*Figure 4—figure supplement 2A,B*). Despite the presence of non-centromeric enrichment in the insoluble chromatin fraction and the lower dynamic range of the CENP-C ChIP data, we called peaks in these two data sets and compared them to the cenH3 peak calls. 460 of the 2060 insoluble chromatin peaks and 163 of the 347 insoluble chromatin peaks coincided with cenH3 peaks. In contrast, only 174 insoluble

chromatin peaks and 26 CENP-C peaks fell within the domains, in both cases fewer sites than expected by chance (p<0.001). Normalized to the genome coverage of domains and peaks, this amounts to an 800-fold enrichment of insoluble chromatin peaks and an almost 2000-fold enrichment of CENP-C peaks at cenH3 peaks compared to cenH3 domains (*Figure 4—figure supplement 2C,D*). These results indicate that the cenH3 peaks identified in this study act as the preferred sites of kinetochore formation.

The precise co-localization of cenH3, CENP-C and insoluble chromatin peaks confirm that these sites correspond to centromeres. Moreover, the number of sites lies in the same order of magnitude as the number of microtubules observed during mitosis, thus providing a mechanistically reasonable alternative to the conundrum of how domains that cover half of the genome can organize a relatively small number of microtubule attachment sites.

## The cenH3 peaks resemble point centromeres

Centromeric nucleosomes in other organisms protect only 80–120 bp of wrapped DNA, compared to 147 bp for canonical nucleosomes (*Dalal et al., 2007*; *Krassovsky et al., 2012*; *Hasson et al., 2013*; *Zhang et al., 2013*), probably due to the reduced wrapping of DNA around them (*Henikoff and Furuyama, 2012*). To examine the size and positioning of the nucleosomes at the 707 centromeric sites, we divided the fragments in the input and cenH3 ChIP samples into size classes of fragments >140 bp representing nucleosomes, and ≤140 bp representing sub-nucleosome-sized particles. In the input sample, we found that two well-positioned nucleosomes flank the cenH3 peaks (*Figure 5A*). These nucleosomes are also visible in modENCODE data for mononucleosomes prepared under native conditions, but not upon formaldehyde-crosslinking, presumably because they become crosslinked to the centromere (*Figure 5—figure supplement 1*). The native input sample also revealed the presence of sub-nucleosome-size fragments over the center of the cenH3 sites, while few of these fragments were found in the flanking regions (*Figure 5A*). In the cenH3 ChIP sample, only relatively few nucleosome-size fragments were recovered, while the majority of the signal came from fragments <140 bp (*Figure 5B*). The insoluble chromatin showed a similar pattern, indicating that the particles bound to DNA at cenH3 sites are similar in the insoluble and in the soluble chromatin fractions (*Figure 5C*). This analysis showed that centromeric sites consist of two well-positioned nucleosomes flanking a single cenH3 nucleosome that wraps less DNA than is wrapped by a canonical nucleosome.

To estimate the size of the DNA associated with these nucleosomes, we plotted the size-distribution of fragments that cross the center of each particle (*Figure 5D*). In the input sample after 2 min of MNase digestion, the fragments that cross the dyad of the flanking nucleosomes show a distribution that peaks at 166 bp, consistent with canonical nucleosomes (*Figure 5E*, black line). The peak lies at 166 bp rather than the 147 bp minimal protected fragment size for mononucleosomes because of the relatively light MNase digestion required to prevent the loss of cenH3 peaks (*Figure 2B*). Progressive MNase digestion reduced the size of the DNA fragments protected by these particles in a manner expected for nucleosomes, analogous to the size pattern observed for bulk chromatin (*Figure 5—figure supplement 2A*). A second peak representing dinucleosomes is also visible at about 330 bp (*Figure 5E*, black line). The fragments that cross the center of the cenH3 nucleosomes in the cenH3 ChIP sample after 2 min of MNase digestion show a strikingly different distribution that is left-shifted and indicate that the nucleosomes that occupy these sites protect only about 60–120 bp (*Figure 5E*, blue line). The fragments in the insoluble chromatin fraction show a very similar distribution, supporting this size estimate (*Figure 5E*, magenta line). The dinucleosome peak in the cenH3 ChIP and insoluble chromatin, representing the centromeric nucleosome and one flanking canonical nucleosome, is equally left-shifted to about 260 bp (*Figure 5E*, blue and magenta lines). These shorter dinucleosome fragments, protected by two neighboring nucleosomes on the same molecule of DNA, can only be explained by the presence of a smaller centromeric nucleosome, because the inferred size estimates are not affected by possible MNase encroachment on the centromeric nucleosome. After 10 min of MNase digestion, the peak of the distribution of MNase fragments that cross the center of the cenH3 sites in the insoluble fractions lies at 66 bp (*Figure 5F*, magenta line), indicating that the minimal protected size of these particles lies in the 60–100 bp range. Moreover, the width at half-height of the average 10 min-digested insoluble chromatin peak that aligns with the cenH3 peaks is 82 bp (*Figure 5—figure supplement 3*). The inferred fragment size protected by the centromeric nucleosome is thus 60–100 bp. These results confirm the findings in other organisms that cenH3 nucleosomes at centromeric sites in *C. elegans* wrap less DNA than that wrapped by canonical nucleosomes.

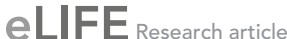

**Figure 5**. CenH3 nucleosomes protect small DNA fragments. (**A, B, C**) Normalized fragment counts in input (**A**), cenH3 ChIP (**B**) and insoluble chromatin (**C**) samples at centromeric sites. MNase fragments were divided into nucleosomal (141–500 bp) and small (21–140 bp) size classes. Average signals within a 1-kb window around all 707 cenH3 sites are plotted. Dashed lines mark the centers of the flanking nucleosomes in (**A**) (black lines) or the centromeric nucleosome in (**B**) (blue line) and in (**C**) (magenta line). (**D**) Cartoon illustrating how MNase fragment size distributions shown in (**E** and **F**) were determined. Fragments that cross the center of the cenH3 nucleosome or of the flanking nucleosomes were counted. (**E** and **F**) MNase fragment size distribution after 2 min (**E**) and 10 min (**F**) of MNase digestion. Input fragments at flanking nucleosomes (black) and cenH3 ChIP fragments (blue) and insoluble chromatin fragments (magenta) at centromeric nucleosomes are shown. Cartoons of the protected particles are shown below each panel. (**G** and **H**) Comparison of worm holocentromere and budding yeast point centromere. (**G**) *C. elegans* holocentromere. Centromere model and cenH3 ChIP over input ratio (all size classes; left y-axis) and nucleosomal signal from input (141–500 bp; right y-axis). Average signals within a 1-kb window around
*Figure 5. Continued on next page*

*Figure 5. Continued*

all 707 cenH3 sites are shown. (**H**) Budding yeast point centromere. Centromere model and data from Krassovsky et al. (**Krassovsky et al., 2012**), cenH3 ChIP over input ratio (left y-axis) and input signal (right y-axis) from all 16 centromeres.

The following figure supplements are available for figure 5:

**Figure supplement 1**. Comparison of input signal at centromeric sites in native and formaldehyde-crosslinked samples.

**Figure supplement 2**. Fragment size distribution for different degrees of MNase digestions.

**Figure supplement 3**. Particle size estimation from overdigested insoluble chromatin.

CenH3 ChIP also revealed mild enrichment over broad domains. Fragment size analysis revealed that the majority of cenH3 nucleosomes in these regions of the genome protect about 135–155 bp of DNA, and that this size distribution is similar in regions between domains (*Figure 5—figure supplement 2B*). This level of protection is consistent with the findings in other organisms that cenH3 can incorporate into canonical-type nucleosomes away from centromeres, in some cases as cenH3-H3.3 heterotypic nucleosomes (*Camahort et al., 2009*; *Krassovsky et al., 2012*; *Lacoste et al., 2014*). This further suggests that the cenH3 domains may be distributed independently of the centromere.

Taken together, fragment size analysis thus revealed that each centromeric site consists of a single cenH3-containing nucleosome that is flanked by two well-positioned canonical nucleosomes (*Figure 5G*). This chromatin landscape is reminiscent of the budding yeast centromere, where a single cenH3 nucleosome assembles on a genetically defined sequence (*Furuyama and Biggins, 2007*; *Henikoff and Henikoff, 2012*). This sequence is flanked by binding sites for centromere-specific protein complexes (Cbf1 and Cbf3) that in turn position two flanking canonical nucleosomes (*Figure 5H*) (*Densmore et al., 1991*; *Krassovsky et al., 2012*). The stable binding of Cbf1 and Cbf3 prevent the centromeric DNA from being occupied by canonical nucleosomes (*Kent et al., 2011*; *Krassovsky et al., 2012*). In *C. elegans*, the flanking nucleosomes are positioned closer together than in yeast, presumably because there are no sequence-specific DNA-binding proteins between the centromeric and the flanking nucleosomes.

Thus, the dispersed centromeric sites in *C. elegans* holocentromeres consist of a cenH3 nucleosome that is associated with 60–100 bp of DNA flanked by two well-positioned nucleosomes, a chromatin pattern with striking similarities to budding yeast point centromeres.

## CenH3 peaks coincide with HOT sites

Budding yeast point centromeres assemble on a genetically defined sequence (*Clarke and Carbon, 1980*). To determine whether the *C. elegans* centromeric sites have common sequence properties, we searched for motifs using MEME (*Bailey et al., 2009*). We found a 15-nt GA-repeat-rich motif that was common to 297 of the 80-bp cores of the centromeric sites (*Figure 6A*). A weaker, but similar GA-rich motif was common to all 707 centromeric sites (*Figure 6—figure supplement 1A*). The 15-nt motif also matched more than 60,000 sites in the genome that are not associated with high cenH3 signal and so is not sufficient to determine cenH3 occupancy.

GAGA-rich sequences are well-characterized targets for the GAGA factor (GAF/Trl) in *D. melanogaster* (*van Steensel et al., 2003*). However, we could not identify a GAF/Trl homologue in the *C. elegans* genome. Instead, the motif we identified is almost identical to the motif associated with *C. elegans* high occupancy target (HOT) sites (*Figure 6A*). These sites were uncovered by the modENCODE consortium through binding-site analysis of 22 transcription factors and operationally defined as sites that are bound by ≥15 transcription factors (*Gerstein et al., 2010*; *Niu et al., 2011*). The sequences at these sites do not contain DNA motifs of known *C. elegans* transcription factors and are therefore expected to bind transcription factors with low affinity. We found that 117 out of 248 HOT sites coincided with cenH3 sites (*Figure 6—figure supplement 1B*; hypergeometric $p=8.6 \times 10^{-161}$). Although this degree of overlap is striking, the actual overlap of cenH3 sites and transcription factor hotspots is likely to be much larger, given the fact that that only 22 transcription factors have been used for the definition of HOT sites, but that there are 934 predicted transcription factors encoded in the *C. elegans* genome (*Reece-Hoyes et al., 2005*; *Gerstein et al., 2010*). We also found that HOT

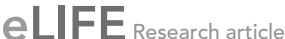

Figure 6. Centromeres coincide with transcription factor hotspots. (**A**) MEME motif for the 80-bp cores of centromeric sites (left) and the 80-bp cores of high occupancy target (HOT) sites (right). (**B**) Heatmaps and average plots of cenH3 ChIP, insoluble chromatin and CENP-C ChIP signal within a 2-kb window around HOT sites, illustrating that HOT sites are highly occupied by cenH3, insoluble chromatin and CENP-C. Each line of the heatmaps represents an individual HOT site. Heatmaps are sorted by decreasing signal.

The following figure supplements are available for figure 6:

**Figure supplement 1**. CenH3 site motif and characterization of HOT sites.

sites show a high signal for cenH3 ChIP, insoluble chromatin, CENP-C ChIP (*Figure 6B*) and well-positioned flanking nucleosomes (*Figure 6—figure supplement 1C*). Moreover, the cenH3 ChIP signal is reduced in KNL-2-depleted animals (*Figure 6—figure supplement 1D,E*). These data suggest that HOT sites and centromeric sites share a similar chromatin landscape and are targeted by both cenH3 nucleosomes and transcription factors.

## Transcription factor occupancy accompanies loss of cenH3 upon exit from the cell cycle

Embryonic cells contain both cenH3 and transcription factors, thus complicating the analysis of the chromatin landscape at centromeric sites. To probe the chromatin landscape in cenH3-depleted cells, we analyzed our previously published data for affinity-purified adult muscle cells, where cenH3 protein is below detection levels (*Figure 7A*), and cenH3 mRNA is significantly depleted (Bayesian t-test; q = 0.0036) (*Steiner et al., 2012*). These samples were MNase-digested >10 min to enrich for mono-nucleosomes (*Figure 2—figure supplement 1*, right panel). Centromeric nucleosomes are unstable under these MNase conditions, and associated fragments are expected to be depleted (*Figure 2*). Despite differences between chromatin preparations from embryos for native ChIP input and whole nuclear DNA extraction from adults for MNase-seq, the overall nucleosome landscapes obtained were very similar (*Figure 7—figure supplement 1A*). In total adult samples, which contain nuclei from dividing germline and embryonic cells, we found a depletion of signal at centromeric sites and well-positioned nucleosomes flanking the sites, reminiscent of the chromatin landscape in embryos (*Figure 7B*, black line). In muscle nuclei, the flanking nucleosomes were also present (*Figure 7B*, red line), and protected a very similar size range of fragments as in the total nuclei sample (*Figure 7C*, left panel). However, the centromeric sites in the muscle nuclei sample were occupied by MNase-stable particles (*Figure 7B*, red line) that protect sub-nucleosome-size fragments (*Figure 7C*, right panel) and likely represent non-nucleosomal DNA-binding proteins.

To test directly whether centromeric sites become occupied by transcription factors upon depletion of cenH3, we profiled the muscle-specific transcription factor HLH-1 in young adults by X-ChIP-seq. HLH-1 is the *C. elegans* myoD homologue and is required for proper myogenesis (*Krause, 1995*). It is exclusively expressed in the same cells that have been used for the muscle chromatin profiling (*Figure 7—figure supplement 1B*). We found that HLH-1 is enriched at the majority of centromeric sites (*Figure 7D*). We also analyzed the previously published transcription factor ChIP-seq datasets for binding at centromeric sites (*Gerstein et al., 2010*). We found that all tested transcription factors profiled in larval instar 2 or later stages, when few somatic cells are still dividing, are enriched at centromeric sites (*Figure 7E*).

These results show that if cenH3 is depleted, centromeric sites are not occupied by canonical nucleosomes, but are bound by sub-nucleosome-size particles at least some of which are known transcription factors. As centromere function is needed only in dividing cells, this possibility is consistent with the observation that some HOT sites have post-mitotic functions as enhancers (*Kvon et al., 2012*; *Chen et al., 2013*).

## Discussion

### Holocentric chromosomes are polycentric

Holocentricity is a common mode of chromosome organization, having evolved from monocentricity at least 13 times, including organisms as diverse as nematodes, moths, and sedges (*Melters et al., 2012*). Based on cytological observations two very different models for holocentricity were proposed more than 60 years ago: diffuse centromeres and polycentromeres (*Schrader, 1947*). We use high resolution mapping of centromeric nucleosomes to demonstrate that the polycentromere model is correct for *C. elegans*, and that *C. elegans* holocentromeres consist of about 100 centromeric sites on each chromosome. The number of discrete centromeric sites is in excess over the observed number of microtubule attachments, which has been estimated to be ~100 for all six chromosomes by electron tomography (*O'Toole et al., 2003*). It is possible that only about 15% of the centromeric sites identified in this study are attached in each mitotic cell, which seems reasonable given that we analyzed a diverse cell population. The availability of multiple centromeric sites might reflect redundancy to assure faithful segregation in every cell cycle, although it is also possible that some sites are rendered inaccessible for cenH3 nucleosomes in some cell lineages due to changes in expression profiles during development.

Polycentromere and diffuse centromere mechanisms might not be mutually exclusive, insofar as a large fraction of cenH3 is incorporated into broad domains that cover half of the genome. It has previously been reported that these domains anti-correlate with transcription (*Gassmann et al., 2012*), and we show that they also anti-correlate with H3.3 and correlate with H3K9me3. These correlations all indicate that cenH3 is preferentially located in regions of low nucleosome turnover, and the intrinsic instability of cenH3 nucleosomes may contribute to losing it from 'open' chromatin (*Conde e Silva,*

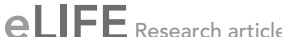

**Figure 7**. Transcription factor occupancy upon loss of cenH3 as cells exit the cell cycle. (**A**) CenH3 in adult germline, intestine and muscle. Staining of a worm section with anti-cenH3, anti-NPP-9 (nuclear pores; staining control) and DAPI are shown. Germ cells in diakinesis are marked with asterisks, muscle cells with arrowheads in the merge. Scale bar is 3 µm. (**B**) Average MNase-seq signal within a 2-kb window around centromeric sites for adult

*Figure 7. Continued on next page*

*Figure 7. Continued*

total and adult muscle nuclei, illustrating that centromeric sites remain occupied in cenH3-depleted cells. MNase digestion >10 min. (**C**) MNase fragment size distribution at flanking nucleosomes and centromeric sites for total adult and adult muscle nuclei, illustrating that at least a sub-population of the particles occupying the centromeric sites are not canonical nucleosomes. (**D**) Heatmap and average plot of HLH-1 ChIP signal from adults within a 2-kb window around cenH3 sites. HLH-1 is a HOT site transcription factor. (**E**) Heatmaps and average plots of ChIP signal for another nine of the transcription factors used to define HOT sites within a 2-kb window around cenH3 sites, data from Gerstein et al. (***Gerstein et al., 2010***). All transcription factors in (**E**) were profiled in the third larval instar except ALR-1 (second larval instar) and PHA-4 (adults). Differences between ChIP and input are plotted in (**D** and **E**). Each line of the heatmaps represents an individual cenH3 site. Heatmaps are sorted by decreasing signal.

The following figure supplements are available for figure 7:

**Figure supplement 1**. Comparison of chromatin from embryos and adults.

*2007*). The domains are in part shaped by transcription in the germline and are present in the early embryo, however, there is no significant RNA Pol II-dependent transcription during the first two rounds of embryonic cell division. In contrast, H3.3 is deposited in the germline and has a well-established role in the inheritance of chromatin states (***Ooi et al., 2006***; ***Ooi et al., 2010***; ***Jullien et al., 2012***). Specifically, H3.3 is retained both in mature sperm and oocytes, suggesting that it transmits epigenetic information through both the maternal and the paternal germline. The maintenance of H3.3 in sperm might explain how the domain pattern is established on paternal chromatin upon fertilization, even though cenH3 is not maintained in mature sperm (***Gassmann et al., 2012***). However, because these domains do not align with the kinetochore, they probably do not have a direct centromere function, although they might serve as cenH3 'reservoirs', in parallel with the suggestion that *Drosophila* transcription factor hotspots might serve as transcription factor reservoirs (***Moorman et al., 2006***).

## Point centromeres are the building block of polycentromeres

We have found that individual centromeric sites resemble budding yeast point centromeres: a single cenH3 nucleosome flanked by two well-positioned nucleosomes. Point centromeres are genetically defined in budding yeast, and satellite repeats help position centromeric nucleosomes at regional centromeres in many species. However, neo-centromeres can form on sequences that are not normally linked to centromeres (***Marshall et al., 2008***; ***Shang et al., 2013***). These observations suggest that centromeric nucleosomes are inherited in a sequence-independent way, so that it might seem surprising that a distinct sequence motif is associated with *C. elegans* centromeric sites. However, given that the motif is short and its abundance in the genome by far exceeds the number of centromeric sites, there is no evidence that it is a direct target for cenH3-nucleosome loading. Rather, the DNA at these sites might disfavor the formation of canonical nucleosomes, allowing centromeric nucleosomes to form in these 'gaps' in the chromatin landscape. A similar model for worm holocentromeres had been proposed by Gu and Fire (***Gu and Fire, 2010***) based on finding ~120-bp 'holes' in the nucleosome landscape that could fit small nucleosomes the size of those previously shown for *Drosophila* cenH3 (***Dalal et al., 2007***).

In *C. elegans*, virtually any piece of DNA injected into the gonad will concatamerize, acquire a centromere and segregate with varying efficiency (***Stinchcomb et al., 1985***; ***Mello et al., 1991***; ***Yuen et al., 2011***). Opportunistic assembly of centromeric nucleosomes at sites of accessible DNA predicts that cenH3 will be loaded onto any fragment of DNA that contains accessible stretches. The fact that new extrachromosomal arrays initially partition passively and only acquire segregation competence after a few cell cycles indicates that centromeric competence can be acquired epigenetically in *C. elegans* (***Yuen et al., 2011***), consistent with an opportunistic gap-filling model.

## A role for transcription factors in holocentromere maintenance?

We found that centromeric sites coincide with HOT sites, which are occupied by many transcription factors without having high binding affinity for any of them. When cells exit the cell cycle and cenH3 is no longer expressed, 'holes' in the chromatin landscape might open up and allow HOT site transcription factors to bind by mass action. Although many HOT site transcription factors are cell type-specific, their non-specific binding to holes vacated by cenH3 nucleosomes would result in high HOT site transcription factor occupancy in multiple cell-types. Indeed, we found that cenH3 is lost in adult muscle cells, but that centromeric sites remain occupied, in part by the muscle-specific transcription factor

HLH-1 and presumably also by other HOT site transcription factors. Virtually all transcription factors profiled in postembryonic tissues (larval instar 2 and later stages) were found at many, if not all centromeric sites. Replacement of cenH3 nucleosomes by transcription factors at HOT sites upon exit from the cell cycle may then result in their reported enhancer activity (*Kvon et al., 2012*; *Chen et al., 2013*).

CenH3 protein has been shown to turn over completely during the mitotic cell cycle, to disappear during the pachytene stage of meiotic prophase, to reappear when nuclei progress into diplotene and to be absent from mature sperm (*Gassmann et al., 2012*). These observations imply that the centromeric sites need to be marked during certain stages of the cell cycle in order to be repopulated at later stages. The coincidence of cenH3 and HOT sites raises the possibility that low-affinity binding of transcription factors by mass action prevents encroachment of nucleosomes and thus, helps to maintain holocentric sites over the course of development.

Our results resolve the long-standing question whether holocentromeres are polycentric or diffuse. We show that *C. elegans* holocentromeres are organized as dispersed point centromeres, consistent with the polycentromere model. Our discovery of the coincidence of centromeric sites with transcription factor hotspots points to a possible mechanism for centromeric site selection and maintenance.

## Materials and methods

### Worm culture and RNAi

We used the standard wild-type strain N2 and OP64 grown at 20°C. Synchronized populations were cultured on peptone-rich plates seeded with *E. coli* strain NA22. To deplete KNL-2 by RNAi, synchronized populations were grown on NA22 until fourth larval instar (L4), washed in M9 buffer and transferred to bacteria expressing dsRNA that targets *knl-2* for 24 hr. Embryos were harvested from adults by sodium hypochlorite treatment.

### Immunofluorescence microscopy

Worms were decapitated with a razor blade in M9 buffer, freeze cracked on dry ice and fixed 2 min in methanol and 4 min in acetone at −20°C. Samples were incubated with anti-cenH3 (rabbit) (*Buchwitz et al., 1999*) and anti-NPP-9 (mouse) (*Sheth et al., 2010*) antibodies overnight at 4°C and with Cy3 donkey anti-rabbit and DyLight 488 donkey anti-mouse antibodies (Jackson ImmunoResearch) for 1 hr at 37°C. Washes were carried out with phosphate buffered saline containing 1% Tween-20 (PBS-T) throughout. Samples were incubated in PBS-T containing 0.01 mg/ml 4′,6-diamidino-2-phenylindole (DAPI) before being mounted. Images were acquired using a Nikon Eclipse 90i microscope (60x lens).

### Native ChIP

N2 embryos were treated in 0.1U/ml chitinase (Sigma) for 30–60 min and washed with buffer A (15 mM Tris–HCl pH7.5, 2 mM $MgCl_2$, 340 mM sucrose, 0.2 mM spermine, 0.5 mM spermidine, 0.5 mM phenylmethanesulfonate [PMSF]). Nuclei were isolated using a glass Dounce homogenizer with 15 strokes each of the loose- and tight-fitting inserts in buffer A supplemented with 0.1% Trition X-100 and 0.25% NP-40 substitute. The homogenate was diluted five times with buffer A, the debris were removed by spinning at 100×*g* for 2 min and nuclei were pelleted by spinning at 1000×*g* for 10 min. Nuclei were transferred to 1 ml 10 mM Tris pH7.5, 2 mM $MgCl_2$, 0.5 mM PMSF and pre-warmed for 5 min at 37°C. $CaCl_2$ to a final concentration of 2 mM, and 0.1 units of micrococcal nuclease (MNase; Sigma–Aldrich) was added. After 1, 2, 5 or 10 min the reaction was stopped by the addition of ethylenediaminetetraacetic acid (EDTA) to a final concentration of 30 mM. A light MNase digestion corresponding to 2 min in this time-course experiment was used for all other experiments unless otherwise noted. Chromatin was solubilized by cavitation using needle extraction (4 times 20 gauge, 4 times 26 gauge), a protocol modified from *Jin and Felsenfeld, 2007*. Soluble chromatin was collected by spinning at 1000×g for 5 min and the supernatant was pooled with additional chromatin solubilized by incubating the pellet in 10 mM Tris pH7.5, 10 mM EDTA, 0.1% Trition X-100, 0.5 mM PMSF for 4 hr at 4°C. The remaining pellet was retained as the insoluble chromatin fraction. Soluble chromatin fractions were combined, NaCl adjusted to 100 mM, debris removed by spinning 4 times at maximum speed for 5 min and pre-cleared by incubation with Dynabeads protein A (Invitrogen). From this input fraction, cenH3 was isolated by incubation with 4 μl anti-cenH3 antibody overnight and protein A dynabeads for 2 hr. Beads were washed three times in 10 mM Tris pH7.5, 100 mM NaCl, 10 mM EDTA,

0.1% Trition X-100, 0.5 mM PMSF and twice in 10 mM Tris pH7.5, 100 mM NaCl, 10 mM EDTA, 0.5 mM PMSF. Chromatin was treated with RNase and Proteinase K, and DNA was isolated with phenol:chloroform and precipitated with ethanol in the presence of glycogen.

## Crosslinked ChIP

For CENP-C ChIP, nuclei were prepared and MNase treated as for native ChIP, except that MNase incubation was done in HM2 (50 mM HEPES pH7.4, 2 mM MgCl$_2$, 0.5 mM PMSF) for 2 min. MNase was inactivated by addition of EGTA to 5 mM and nuclei were washed once in HM2. Chromatin was crosslinked in 1% formaldehyde for 10 min. Crosslinking was quenched by adding glycine to 125 mM for 10 min. Nuclei were washed with HM2 and lysed in 50 mM Tris pH7.5, 10 mM EDTA, 1% SDS, 0.5 mM PMSF by vortexing for 2 min. The lysate was diluted to 20 mM Tris pH7.5, 150 mM NaCl, 0.1% SDS, 1% Triton X-100, 2 mM EDTA, 0.5 mM PMSF and sonicated with a Sonic Dismembrator Model 500 (Fisher Scientific) for 40s at 30% amplitude. Debris removal, pre-clearing and CENP-C ChIP were done as for native ChIP, with the antibody from *Moore and Roth, 2001*. Beads were washed twice with 20 mM Tris pH7.5, 150 mM NaCl, 0.1% SDS, 1% Triton X-100, 2 mM EDTA, once each in 20 mM Tris pH7.5, 500 mM NaCl, 0.1% SDS, 1% Triton X-100, 2 mM EDTA and 20 mM Tris pH7.5, 250 mM LiCl, 1% sodium deoxycholate, 1% NP-40 substitute, 2 mM EDTA and once in TE. Crosslinks were reversed overnight at 65°C, chromatin was treated with RNase and Proteinase K, and DNA was isolated with phenol:chloroform and precipitated with ethanol in the presence of glycogen.

For HLH-1 ChIP, OP64 worms were washed in PBS, ground under liquid nitrogen and resuspended in PBS containing 1x Complete Protease Inhibitor Cocktail (Roche). Proteins were crosslinked with 1% formaldehyde for 15 min, the reaction was quenched with 125 mM glycine for 10 min, the volume increased to 50 ml, and chromatin pelleted by spinning at 2000×g for 10 min. The pellet was washed again in PBS, resuspended in 50 mM Tris pH7.5, 10 mM EDTA, 1% SDS containing 1x Complete Protease Inhibitor Cocktail (Roche), incubated 10 min at room temperature, diluted to 20 mM Tris pH7.5, 150 mM NaCl, 0.1% SDS, 1% Triton X-100, 2 mM EDTA containing 1x Complete Protease Inhibitor Cocktail (Roche) and sonicated with a Sonic Dismembrator Model 500 (Fisher Scientific) for 4 min at 30% amplitude. Debris was removed by spinning twice at maximum speed for 5 min. The extract was incubated with an anti-FLAG M2 antibody (Sigma) overnight at 4°C. Protein G beads pre-blocked with BSA and yeast tRNA were added for 4h. Beads were washed and DNA isolated as described for CENP-C ChIP above.

## Illumina sequencing and data analysis

Libraries were prepared using a modified Illumina paired-end library protocol as described in *Henikoff et al., 2011*. Cluster generation, followed by 25 rounds of paired-end sequencing in an Illumina Hi-Seq 2000, was performed by the FHCRC Genomics Shared Resource.

After processing and base-calling by Illumina software, paired-end reads were mapped to the *C. elegans* genome release WS220 using Novoalign (http://www.novocraft.com) with default parameters, except that each multiple hit was mapped to one site chosen at random (Novoalign parameter -r Random). The number of inserts aligned to each 10-bp interval of the genome was counted, and the interval counts were normalized by dividing by the total number of counts for all intervals, and then scaled by multiplying by the number of bases in the genome. We considered fragments >140 bp to represent nucleosomes, the in silico equivalent of excising a gel slice around the ~150-bp size range from an MNase-digested chromatin ladder and extracting the DNA for single-end sequencing. As we use a modified paired-end sequencing protocol to include all fragments >25 bp (*Henikoff et al., 2011*), we can accomplish the size cut more precisely by mapping only the reads in the nucleosome size range. Simple repeat regions were downloaded from www.wormbase.org and excluded from all analyses.

To call peaks, given the discrete nature of the sites of high cenH3 signal, we set a threshold and considered all the features with higher counts as peaks. For cenH3 peak calling, input counts were subtracted from cenH3 ChIP counts for two biological replicates (2 min MNase), and peaks that exceeded 30 counts in at least one of the biological replicates were considered positive. For CENP-C peak calling, input counts were subtracted from CENP-C ChIP counts, and peaks that exceeded 20 counts were considered positive. For insoluble chromatin peak calling, peaks that exceeded 100 counts were considered positive.

To normalize against input, we used log$_2$-ratios only to compare to previously published array data. Otherwise, we consider input reads a separate "blank" experiment that we subtract from the ChIP counts.

## Acknowledgements

We thank Jorja Henikoff for data analysis, Christine Codomo, Terri Bryson and Aaron Hernandez for reagent preparation, Srinivas Ramachandran, Sivakanthan Kasinathan and Christopher Weber for help with the data analysis, James Priess for providing reagents and lab space, Mark Roth and Mike Morrison for providing antibodies and Sue Biggins, Takehito Furuyama, James Priess, Peter Skene, Paul Talbert, Christopher Weber and Gabriel Zentner for comments on the manuscript. Some strains were provided by the CGC, which is funded by NIH Office of Research Infrastructure Programs (P40 OD010440). Sequence data are in the Gene Expression Omnibus (GEO) database under accession number GSE44412.

## Additional information

### Funding

| Funder | Grant reference number | Author |
|---|---|---|
| Howard Hughes Medical Institute | Henikoff | Florian A Steiner, Steven Henikoff |
| Swiss National Science Foundation | PBSKP3-124362 | Florian A Steiner |
| National Institutes of Health | U01-HG004274 | Florian A Steiner, Steven Henikoff |

The funders had no role in study design, data collection and interpretation, or the decision to submit the work for publication.

### Author contributions

FAS, Conception and design, Acquisition of data, Analysis and interpretation of data, Drafting or revising the article; SH, Conception and design, Analysis and interpretation of data, Drafting or revising the article

## Additional files

### Major dataset

The following dataset was generated:

| Author(s) | Year | Dataset title | Dataset ID and/or URL | Database, license, and accessibility information |
|---|---|---|---|---|
| Steiner FA, Henikoff S | 2013 | Holocentromeres are dispersed point centromeres localized at transcription factor hotspots | GSE44412; http://www.ncbi.nlm.nih.gov/geo/query/acc.cgi?acc=GSE44412 | Publicly available at the Gene Expression Omnibus (http://www.ncbi.nlm.nih.gov/geo/). |

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
