## [Decision Letter]

Thank you for sending your work entitled “Holocentromeres are dispersed point centromeres localized at transcription factor hotspots” for consideration at *eLife*. Your article has been favorably evaluated by a Senior editor, Detlef Weigel, and 3 reviewers, one of whom, Asifa Akhtar, is a member of our Board of Reviewing Editors.

The Reviewing editor and the other reviewers discussed their comments before we reached this decision, and the Reviewing editor has assembled the following comments to help you prepare a revised submission.

This report describes the distribution of centromeric chromatin in *C. elegans* and proposes a new model for “holocentric” chromosomes. A previous study had mapped the centromeric histone cenH3 by ChIP-chip to large domains throughout the genome and concluded that the centromeres of C. elegans do not reside in discrete locations but form randomly in intergenic regions (“diffuse centromeres”). This study uses a different method (ChIP-Seq) to map cenH3 and comes to a very different conclusion: in addition to the broad domains, they identify ∼700 discrete locations for cenH3 and conclude that the centromeres of *C. elegans* are in fact similar to the point centromeres of yeast. The authors also describe that the centromeres correspond to the so-called “HOT” sites where transcription factors tend to accumulate in non-dividing cells. These observations are novel, resolve a long-standing question regarding the structure of holocentric chromosomes, and have intriguing implications for the formation and maintenance of centromeres.

All three reviewers are generally interested in your work and find the study of significant importance to be suitable for publication in *eLife*. However, the consensus among the reviewers is that the following aspects of the paper must be strengthened before we proceed further. Here are the most important points that should be addressed upon revision.

The authors should demonstrate more convincingly that cenH3 peaks and not broad cenH3 domains are the centromeres that build the kinetochore. To be more convincing they need to present the CENP-C X-chip in a more transparent and comprehensive manner:

1) In Figure 4, only two regions of the CENP-C X-Chip are shown. To assess the quality of the CENP-C X-Chip it would be really important to see a larger region of DNA as it has been done in other figures of this paper.

- How many CENP-C sites are identified?

- What sub-fraction of cenH3 peaks overlaps with what subfraction of the other sequences with centromere-like behavior: insoluble sites and CENP-C peaks (a Venn diagram or other graphical means could be useful). Most importantly, how does this compare to broad cenH3 domains?

2) Given that the MNase digest is important to the argument that the identified cenH3 sites show the properties of point centromeres, the authors should show a control in Figure 2 for MNase digestion, such as an H3 nucleosome where the amplitude is less sensitive when compared to cenH3.

3) More information on the peak calling should be provided as this confused two reviewers.

---

## [Author Response]

*1) In*
Figure 4*, only two regions of the CENP-C X-Chip are shown. To assess the quality of the CENP-C X-Chip it would be really important to see a larger region of DNA as it has been done in other figures of this paper*.

The quality of the CENP-C X-ChIP should be most evident from the heat map in Figure 4, which depicts the signals over the 707 centromere sites. We did not use the CENP-C data to call sites; only to confirm that the sites that we called based on cenH3 data were enriched in CENP-C, and the Figure 4 heat map and Figure 4—figure supplement 2 boxplots clearly make this point. It shows that the enrichment of CENP-C at cenH3 sites is not restricted to the two example peaks shown and extends to the large majority of cenH3 sites genome-wide. We acknowledge that the CENP-C X-ChIP data are inherently much noisier than the cenH3 native ChIP data. The reason for this is at least in part biological, as cenH3 is present on chromatin through the vast majority of the cell cycle, while CENP-C was previously shown to be localized to cenH3 sites during mitosis, but not during interphase (Moore and Roth, 2001 PMID:11402064). As only a small fraction of the embryonic cells analyzed are in mitosis, the signal-to-noise ratio is inevitably much lower. Nevertheless, CENP-C ChIP signal-to-noise is sufficient to confirm that the cenH3 sites correspond to kinetochore sites, and we have further strengthened this conclusion as described below.

*- How many CENP-C sites are identified*?

*- What sub-fraction of cenH3 peaks overlaps with what subfraction of the other sequences with centromere-like behavior: insoluble sites and CENP-C peaks (a Venn diagram or other graphical means could be useful). Most importantly, how does this compare to broad cenH3 domains*?

We agree that a direct head-to-head comparison of cenH3 peaks and broad domains with respect to insoluble sites and CENP-C peaks is important to strengthen our conclusion that cenH3 sites correspond to kinetochore sites. There are 347 CENP-C peaks, of which 163 overlap with the cenH3 sites, whereas only 26 CENP-C peaks overlap with the domains. Based on the fact that the coverage of cenH3 peaks is only 0.3% that of broad domains, the enrichment of CENP-C at cenH3 peaks relative to cenH3 domains is (163/26)/0.003 ≈ 2000-fold. Similarly, there are 2060 peaks in the insoluble chromatin, 460 of which coincide with the cenH3 peaks, compared to 147 for the domains, which implies a relative enrichment of insoluble chromatin at cenH3 peaks to cenH3 broad domains of (147/460)/0.003 ≈ 800-fold. Importantly, the densities of both CENP-C and insoluble chromatin peaks within the domains are significantly *lower* than within the rest of the genome. That is, our CENP-C ChIP data not only confirm that the cenH3 peaks are kinetochore sites, the data also argue against the hypothesis that broad domains harbor kinetochore sites. We have included these new data in Figure 4—figure supplement 2, panels C and D and now make this point in the text (Results section entitled “CenH3 peaks correspond to kinetochore sites”).

*2) Given that the MNase digest is important to the argument that the identified cenH3 sites show the properties of point centromeres, the authors should show a control in*
Figure 2
*for MNase digestion, such as an H3 nucleosome where the amplitude is less sensitive when compared to cenH3*.

We agree that we had not sufficiently described the characteristic MNase sensitivity of the point centromere sites. To strengthen this point, we have substantially revised Figure 2, added Figure 2—figure supplement 2, and expanded the text describing the MNase sensitivity (Results section entitled “cenH3 peaks are hyper-sensitive to MNase digestion”). We now show the progression of the MNase digest separately for the input and cenH3 ChIP samples, both for an extended region around the two example peaks (Figure 2) and at all sites genome-wide (Figure 2). This illustrates that the cenH3 peaks are sensitive to MNase, while the surrounding chromatin features remain relatively unaffected. For comparison, we now also include the pattern for progressive digestion of the input chromatin at the +1 nucleosomes of genes (Figure 2).

To quantify the MNase-sensitivity of cenH3 nucleosomes, the nucleosomes flanking the cenH3 peaks and the +1 nucleosomes, we plotted the occupancy at these features relative to the occupancy of first time point. This clearly shows that the cenH3 peaks are more MNase-sensitive than both the flanking and +1 nucleosomes (Figure 2).

*3) More information on the peak calling should be provided as this confused two reviewers*.

We have extended the description of our very simple peak calling in the Methods section (penultimate paragraph of the section entitled “Illumina sequencing and data analysis”) and added a sentence in the Results section (entitled “High resolution mapping of cenH3 reveals discrete high occupancy sites”). We classified all sites that exceeded 30 normalized counts in at least one of two biological replicate (cenH3 ChIP minus input) as peaks. 30 counts is the equivalent of the genome-wide mean plus seven standard deviations.